

# Has_circRNA_0122683 (circ-PRKCI) relieves ferroptosis of HPAEpiCs in sepsis-induced acute lung injury by sponging miR-382-5p

Limei Yan[1], Xiajun Lu[1], Ning Wang[1] and Peng Jia[2]

[1] Intensive Care Medicine Department, The People's Hospital of Suzhou New District, Suzhou, China
[2] Trauma Orthopedics, The People's Hospital of Suzhou New District, Suzhou, China

Corresponding author
Peng Jia, jiapeng888@126.com

## ABSTRACT

Circular ribonucleic acid (RNA) protein kinase C iota (circ-PRKCI, hsa_circRNA_0122683) has been previously reported to be involved in the development of sepsis. However, the knowledge regarding the potential role and mechanism of circ-PRKCI in sepsis-induced acute lung injury (ALI) is unclear. An *in vitro* cellular model of sepsis-ALI was simulated by the treatment of lipopolysaccharide (LPS) in human pulmonary alveolar epithelial cells (HPAEpiCs). The expression of circ-PRKCI in plasma samples from sepsis patients with or without ALI as well as sepsis-ALI cell model was determined by quantitative real-time PCR (qRT-PCR). The diagnostic utility of circ-PRKCI was analyzed using receiver operating characteristic (ROC) curves. The levels of iron content ($Fe^{2+}$), glutathione (GSH), malondialdehyde (MDA), and reactive oxygen species (ROS) were detected using corresponding commercial kits. The assessment of cell viability and production of pro-infammatory cytokines (IL-6, IL-1β and TNF-α) was measured using Cell Counting Kit-8 (CCK-8) and enzyme-linked immunosorbent assay (ELISA). The targeting relationship between circ-PRKCI and miR-382-5p was predicted by bioinformatics analysis, and subsequently confirmed by luciferase reporter and RNA immunoprecipitation (RIP) assays. Results shows that decreased circ-PRKCI expression but increased miR-382-5p expression was observed in sepsis patients with ALI and sepsis-induced ALI cell model. The area under the curve values of ROC curves for circ-PRKCI in differentiating septic ALI patients from healthy individuals and septic non-ALI patients were 0.996 and 0.999, respectively. Functional *in vitro* assays revealed that enforced expression of circ-PRKCI alleviated LPS-induced ferroptosis and inflammatory response of HPAEpiCs, which were reversed by Erastin or FIN56 administration. Mechanistically, circ-PRKCI was identified as a sponge of miR-382-5p and negatively regulated miR-382-5p expression. Further rescue experiments showed that miR-382-5p overexpression could compromise the anti-ferroptosis and anti-inflammatory response effects of circ-PRKCI on LPS-induced injury of HPAEpiCs. Our study demonstrated that circ-PRKCI may be a promising biomarker for septic ALI diagnosis. circ-PRKCI inhibits ferroptosis and inflammatory response in sepsis-induced ALI by sponging miR-382-5p, indicating

that the circ-PRKCI/miR-382-5p axis might be a novel therapeutic target for the treatment of sepsis-induced ALI.

## INTRODUCTION

Sepsis is a life-threatening organ dysfunction that originates from the abnormal response of the body to severe trauma, burn, shock, infection, and surgery (*Cecconi et al., 2018*). Acute lung injury (ALI) is one of the common complications of sepsis that is characterized by rapid-onset respiratory failure due to dysfunction of epithelial permeability and microvascular leakage, which causes high morbidity and even death, and has become a major public health burden worldwide (*Xu et al., 2024*). Although the intensive advances in symptomatic and supportive treatment such as advanced antibiotics, neuromuscular blocking agents (NMBAs), high-frequency oscillatory ventilation (HFOV), and extracorporeal membrane oxygenation (ECMO), annual mortality caused by ALI remains a serious issue (*Mokrá, 2020*). Thus, efforts to develop new targets for treatment of ALI are an attractive endeavor. A growing number of studies in the past few years have manifested that active oxidative stress and inflammatory response lead to pathological progression of ALI (*Bezerra et al., 2023*). Recent studies suggested that that ferroptosis, an iron-dependent new programmed cell death, has been observed in ALI, and prevention of ferroptosis is effective for improving ALI treatment (*Liu, Zhang & Xie, 2022*; *Wang, Zhao & Xiao, 2023*). A growing number of meta-analysis studies have summarized the roles and regulatory mechanisms of ferroptosis in different tumors (*Hang, Du & Song, 2024*; *Li et al., 2024*; *Park et al., 2021*). Nevertheless, the regulators of ferroptosis in ALI remain largely unknown.

As one of the endogenous non-coding ribonucleic acids (RNAs), circular RNAs (circRNAs) have stable covalent closed loop structures by back splicing, including exonic circRNAs, intron circRNAs, and exon-intron-derived or retained intron circRNAs. It is reported that approximately 450 circRNAs are uniquely enriched in human lung tissues, compared with other tissues by RNA sequencing analysis, and some of these circRNAs have been reported to play crucial roles in human lung development (*Tong et al., 2023*). Mounting evidence indicated that dysregulation of circRNAs has been related to various respiratory system diseases, and targeting circRNAs may be a novel therapeutic strategy (*Wang et al., 2019*). Recently, the regulatory roles of circRNAs in the pathogenesis of ALI have been gradually illustrated (*Gao et al., 2024*). For example, circ_0114428 is upregulated in sepsis patients and circ_0114428 knockdown protects against sepsis-associated ALI in lipopolysaccharide (LPS)-induced human pulmonary alveolar epithelial cells (*Zhao, Zhao & Duan, 2024*). *You & Zhang, 2022* demonstrated that circ-WDR33 expression is low in sepsis-ALI patients and alleviate LPS-induced inflammatory and apoptotic injury, permeability enhancement and tubule formation arrest in human pulmonary microvascular endothelial cells. As for circ-C3P1, *Jiang et al. (2020)* disclosed that

overexpression of circ-C3P1 in sepsis-induced ALI mice attenuates pulmonary injury, pro-inflammatory cytokine production and cell apoptosis. However, the functions of most circRNAs are still not fully analyzed and still need to be explored.

Circular RNA protein kinase C iota (Circ-PRKCI) (hsa_circRNA_0122683) is located at the chr3:170011170-170016898 position of the human genome, originating from exons 14, 15, 16 and 17 of its host gene protein kinase C iota (PRKCI, NM_002740), which has been reported to be aberrantly expressed in various human cancers such as hepatocellular carcinoma (*Chen et al., 2021*), papillary thyroid cancer (*Liu et al., 2021*), and renal cell carcinoma (*Qian et al., 2022*). Additionally, *Wei & Yu (2020)* explored the clinical value of circ-PRKCI in sepsis patients, *Xiong et al. (2021)* focused on the effect of circ-PRKCI in lipopolysaccharide (LPS)-mediated HK2 cell injury, and these findings suggested that circ-PRKCI expression is downregulated in sepsis patients and could alleviate sepsis-induced acute kidney injury. However, it is unclear if circ-PRKCI would affect the development of sepsis-induced ALI. Therefore, in the present study, we aimed to identify the biological function of circ-PRKCI *in vitro* cellular model of sepsis-ALI by LPS administration. Preliminary bioinformatics analysis revealed that circ-PRKCI complementarily paired with miR-382-5p, suggesting that circ-PRKCI may act as competing endogenous RNA (ceRNA) to regulate miR-382-5p expression. Importantly, the potential regulatory mechanism between circ-PRKCI and miR-382-5p was also investigated. Taken together, our study identifies circ-PRKCI as a potentially novel diagnostic biomarker and therapeutic target for sepsis-induced ALI.

## MATERIALS AND METHODS

### Plasma samples

This study was a retrospective research, the protocols and procedures of clinical samples related experiments were strictly reviewed and permitted by the Ethics Committee of The People's Hospital of Suzhou New District following the Helsinki Declaration (No. 2024-110), and written informed consent was obtained from all participants or their delegates before the beginning of this study. The venous blood samples (5 mL) were collected from 80 sepsis patients (included 36 women and 44 men with an average age of 46.17 years) and 53 age and gender-matched healthy individuals (included 21 women and 32 men with an average age of 44.90 years) recruited between May 2022 and May 2024 from The People's Hospital of Suzhou New District (Suzhou, China). The 80 sepsis patients all had bacterial infections, including 31 septic ALI patients and 49 septic non-ALI patients. Patients who transferred from other hospitals, received immunosuppressive therapy, pregnant or lactating were excluded. Sepsis and sepsis-induced ALI were diagnosed according to the corresponding international criteria (*Singer et al., 2016*; *Raghavendran & Napolitano, 2011*). Healthy individuals had no history of sepsis or other severe infection or systemic diseases. All plasma samples were obtained from peripheral venous blood by centrifugation at 3,000× *g* for 10 min at 37 °C, and were immediately frozen in liquid nitrogen and subsequently stored at −80 °C until RNA extraction.

## Cell culture and LPS exposure

Human pulmonary alveolar epithelial cells (HPAEpiCs) were acquired from Shanghai Honsun Biological Technology Co., Ltd. (Shanghai, Beijing, China), and maintained in Dulbecco's Modified Eagle Medium (DMEM; Life Technologies, Carlsbad, CA, USA) containing 10% fetal bovine serum (FBS; Gibco, Waltham, MA, USA) and 1% penicillin/streptomycin (Sigma-Aldrich, St. Louis, MO, USA). The cell culture condition was 5% $CO_2$ and 95% air at a constant temperature of 37 °C. To construct sepsis-ALI cell model, HPAEpiCs were treated with 1 µg/mL of LPS (Beyotime Biotechnology, Shanghai, China) for different times (3, 6, 12, or 24 h), as previously described (*Zhang et al., 2021a*).

## Cell transfection

To overexpress circ-PRKCI or miR-382-5p, circ-PRKCI overexpression vector (circ-PRKCI) and the corresponding control vector (circ-NC), miR-382-5p mimic (5′-GAAGUUGUUCGUGGUGGAUUCG-3′) and mimic negative control (mimic NC; 5′-GGUUCGUACGUACACUGUUCA-3′), were synthesized from Guangzhou Ribobio Co., Ltd. (Guangzhou, China). When cell confluence reached ~75%, HPAEpiCs were transfected with the above plasmids or oligonucleotides using Lipofectamine 3000 reagent (Invitrogen, Waltham, MA, USA) in accordance with the manufacturer's manual. A total of 48 h after transfection, the efficiency of the transfection was verified by quantitative real-time PCR (qRT-PCR). After the treatment with 5.0 mmol/L of Erastin or 4.0 µmol/L of FIN56 (ferroptosis inducer; Selleck Chemicals, Houston, TX, USA), the transfected HPAEpiCs were exposed to 1 µg/mL of LPS for further functional experiments.

## RNA extraction and qRT-PCR

Total RNA was extracted from plasma samples and cells using TRIZOL reagent (TaKaRa, Kusatsu, Japan), and then quantified by NanoDrop 2000/2000c (Thermo Scientific, Waltham, MA, USA). OD260/OD230 ratios >1.8 but <2.1 were accepted. For synthesized complementary DNA (cDNA) of miRNAs, the RNA was reversely transcribed with the miScript II RT Kit (QIAGEN) according to the manufacturer's protocols. For cDNA of circRNAs and mRNAs, the RNA was reversely transcribed with using the PrimeScript RT Reagent Kit (TaKaRa, Kusatsu, Japan) in accordance with the manufacturer's manual. qRT-PCR reaction was performed with the miScript SYBR Green PCR Kit (QIAGEN) or SYBR® Premix Ex Taq™ II reagent kit (TaKaRa, Kusatsu, Japan) on an ABI 7900HT Real-Time PCR System (Applied Biosystems, Waltham, MA, USA) according to the manufacturer's instructions, with amplification condition at 95 °C for 15 min followed by 40 cycles of 95 °C for 10 s and 59 °C for 30 s and 45 s at 72 °C. The primers used in this study were synthesized from Shanghai Sangon Biotech Co., Ltd (Shanghai, Beijing, China), and the specific sequences were shown in Table S1. Glyceraldehyde-3-phosphate dehydrogenase (GAPDH) was used as the standardized internal reference of circ-PRKCI and mRNAs, and U6 was utilized as the standardized internal control of miRNAs, and relative expression level of each gene was calculated using the $2^{-\Delta\Delta Ct}$ method.

## Ribonuclease R (RNase R) and Actinomycin D (ActD) treatment

For Ribonuclease R (RNase R) treatment, 3 μg of total RNA from HPAEpiCs was treated with 2 U/μg RNase R (Epicenter Biotechnologies) at 37 °C for 15 min to remove the linear RNA. For Actinomycin D (ActD) treatment, when cell confluence reached ~75%, HPAEpiCs were treated with 2.0 μg/mL ActD (MedChem Express, Monmouth Junction, NJ, USA) and collected at different times (0, 4, 8, 12, or 24 h). Afterwards, the RNA was collected, and stability of circ-PRKCI and PRKCI mRNA was evaluated by qRT-PCR.

## Luciferase reporter assay

The sequences of circ-PRKCI were obtained from circBase (http://www.circbase.org), and miR-382-5p sequences were derived from miRBase (https://www.mirbase.org/). We predicted the miRNA binding sites of circ-PRKCI using the bioinformatics databases CircInteractome (https://circinteractome.nia.nih.gov/index.html) and circBank1.0 (http://www.circbank.cn/index.html). The luciferase reporter assay was employed to confirm the target binding sites between miR-382-5p and circ-PRKCI. The wild-type (wt) circ-PRKCI sequence or mutant (mut) sequence was inserted into the 3′-UTR downstream region of Renilla luciferase in the psiCHECKTM-2 vector (Promega) to construct corresponding luciferase reporter plasmids. When cell confluence reached ~75%, HPAEpiCs were co-transfected with miR-382-5p mimic or mimic NC and either wt-circ-PRKCI plasmid or mut-circ-PRKCI plasmid or basic plasmid with Lipofectamine 3000 reagent. A luciferase reporter assay kit (Promega) was used to determine the Renilla and Firefly luciferase activity. The relative luciferase activity was shown as Renilla luciferase activity normalized to that of Firefly.

## RNA immunoprecipitation (RIP) assay

The Magna RIP Kit (Millipore) was applied for RNA immunoprecipitation (RIP) assay according to the instruction provided by the manufacturer. In brief, magnetic beads were incubated with argonaute2 (Ago2) antibody (Millipore) or corresponding immunoglobulin G (IgG) antibody (Millipore) overnight at 4 °C. Then the beads were incubated with cell lysates from HPAEpiCs, and treated with proteinase K to digest proteins. After the elution of precipitated complex from beads, the immunoprecipitated RNAs were isolated and purified for qRT-PCR analysis. Relative enrichment of miR-382-5p and circ-PRKCI was computed with the normalization to the input.

## Detection of cell viability

The viability of cells was examined by Cell Counting Kit-8 (CCK-8) assay according to the manufacturer's instructions. HPAEpiCs underwent various treatments were seeded in a 96-well plate at a density of 5,000 cells per well and incubated for 24 h. Then, 10 μL of CCK-8 reagent (Beyotime Biotechnology, Shanghai, China) was added into cell culture medium on the indicated time and incubated at 37 °C for 3 h. Finally, cell viability ability was assessed by measuring the absorbance at wavelength of 450 nm using an ultraviolet spectrophotometer (Thermo Scientific, Waltham, MA, USA).

## Assessment of pro-infammatory cytokines

The production of pro-infammatory cytokines, including interleukin 6 (IL-6), interleukin 1 beta (IL-1β) and tumor necrosis factor alpha (TNF-α), was measured using enzyme-linked immunosorbent assay (ELISA, Helsinki, Finland). The cell supernatants from HPAEpiCs that underwent various treatments were collected and tested with the corresponding ELISA kits (R&D Systems, Minneapolis, MN, USA) for IL-6, IL-1β and TNF-α according to the manufacturer instructions.

## Measurement of iron content ($Fe^{2+}$)

The $Fe^{2+}$ level was measured by the Iron Assay Kit (Abcam). Following the guidelines, HPAEpiCs that underwent various treatments were plated in six-well plates at a density of $3 \times 10^6$ cells per well for 24 h. After that, the cells were lysed and cell supernatants were collected by centrifugation at $13,000\times g$ for 15 min at 4 °C. The supernatants were treated with an iron reducer for 30 min at 37 °C, followed by incubation with iron probe for 60 min at 37 °C in the dark. Finally, absorbance was immediately measured with an ultraviolet spectrophotometer at a wavelength of 593 nm.

## Glutathione (GSH) assay

The glutathione (GSH) level was detected using the GSH detection kit (Beyotime Biotechnology) according to the manufacturer's instructions. Briefly, HPAEpiCs that underwent various treatments were seeded into 24-well at density of $1 \times 10^5$ cells per well for 24 h. After extraction and centrifugation, the release of GSH in the supernatants was detected using an ultraviolet spectrophotometer at wavelength of 412 nm.

## Malondialdehyde (MDA) assay

The malondialdehyde (MDA) concentration was evaluated by the Lipid Peroxidation MDA Detection Kit (Abcam) according to the manufacturer's protocols. Briefly, HPAEpiCs that underwent various treatments were cultured for 24 h and treated with MDA lysis buffer. Afterward, the cell supernatants were mixed with thiobarbituric acid (TBA) to form an MDA-TBA adduct. With the aid of an ultraviolet spectrophotometer, the absorbance of the MDA-TBA adduct was detected at a wavelength of 532 nm.

## Reactive oxygen species (ROS) assay

The Cellular ROS Assay Kit with by H2DCFH-DA fluorescent probe (Abcam, Cambridge, UK) was applied for reactive oxygen species (ROS) detection. HPAEpiCs that underwent various treatments were plated in six-well plates at density of $3 \times 10^6$ cells per well for 24 h, and incubated with 10 µmol/L of H2DCFH-DA for 30 min at 37 °C, away from light. After washing with phosphate buffered saline (PBS), images were observed and photographed under a fluorescence microscope (Leica, Wetzlar, Germany), and fluorescence strength was computed with Image J software (National Institutes of Health, Bethesda, MD, USA). All experimental procedures were carried out in strict accordance with the procedures specified in the instructions.

## STATISTICAL ANALYSIS

Statistical analysis was analyzed with the Statistical Product and Service Solutions (SPSS) version 24.0 (SPSS). Each experiment was repeated at least three times, data were exhibited as the mean ± standard deviation (SD) of the average of experiments. The differences between two groups were compared by Student's t-test. The differences for three groups or more groups were analyzed by one-way analysis of variance (ANOVA) followed by *Post-Hoc* test. Receiver operating characteristic (ROC) curves were constructed and the area under the curve (AUC) value was considered to assess diagnostic utility of circ-PRKCI. The value of $P < 0.05$ was considered to be statistically significant.

## RESULTS

### Circ-PRKCI is notably diminished in sepsis-induced ALI

First, to validate the expression pattern of circ-PRKCI in sepsis-induced ALI, we examined the circ-PRKCI expression in plasma samples from 31 septic ALI patients and 49 septic non-ALI patients, as well as 53 healthy individuals. As shown in Fig. 1A, circ-PRKCI was lowly expressed in septic ALI patients compared with healthy controls. Meanwhile, the expression of circ-PRKCI was lower in septic ALI patients than that in septic non-ALI patients. Subsequently, HPAEpiCs were exposed to 1 µg/mL LPS for different times, we found that circ-PRKCI expression was significantly reduced in LPS-treated cells compared to untreated cells, and circ-PRKCI was progressively reduced in a time-dependent manner (Fig. 1B). In addition, we evaluated the stability of circ-PRKCI in HPAEpiCs. After treatment with RNase R, circ-PRKCI was more stable than the linear PRKCI mRNA (Fig. 1C). Also, HPAEpiCs treatment with ActD, which can block new transcription, revealed that circ-PRKCI was more stable in comparison to PRKCI mRNA (Fig. 1D). These results collectively demonstrated that circ-PRKCI is a stable circRNA generated from PRKCI, and is substantially downregulated in sepsis-induced ALI.

### The diagnostic performance of circ-PRKCI in sepsis-induced ALI

The potential diagnostic value of circ-PRKCI in sepsis-induced ALI was evaluated through ROC curves, as shown in Fig. 2A, the AUC value for circ-PRKCI in differentiating septic ALI patients from healthy individuals was 0.996 (95% confidence interval (CI) [0.987–1.000]), and the sensitivity and specificity were 96.77% and 100.00%. What is more, circ-PRKCI could discriminate septic ALI patients from septic non-ALI patients with an AUC value of 0.999 (95% CI [0.997–1.000]), sensitivity was 100.00%, and specificity was 97.96% (Fig. 2B). These results indicated that circ-PRKCI may serve as a promising biomarker for clinical diagnosis of sepsis-induced ALI.

### Circ-PRKCI ameliorates LPS-induced ferroptosis and inflammatory response of HPAEpiCs

In order to study the possible function of circ-PRKCI on sepsis-induced ALI, we applied circ-PRKCI overexpression vector to enhance endogenous expression of circ-PRKCI in HPAEpiCs. The efficiency for circ-PRKCI overexpression was validated by qRT-PCR, and
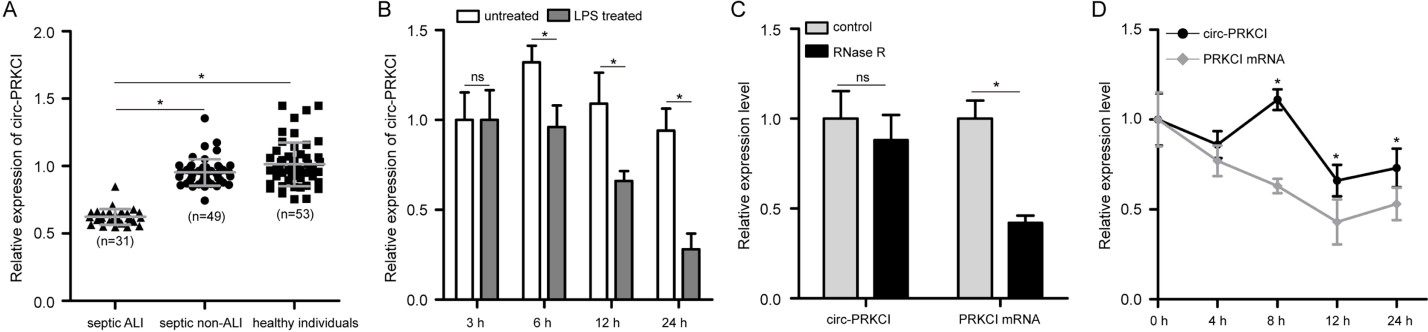

**Figure 1 The expression of circ-PRKCI in sepsis-induced ALI.** (A) The expression levels of circ-PRKCI in plasma samples from 31 septic ALI patients, 49 septic non-ALI patients and 53 healthy individuals were measured by qRT-PCR. (B) The circ-PRKCI expression in HPAEpiCs treated with 1 μg/mL LPS for different times by qRT-PCR. (C) The relative levels of circ-PRKCI and PRKCI mRNA were analyzed by qRT-PCR in HPAEpiCs treated with or without RNase R. (D) qRT-PCR was conducted to determine the relative levels of circ-PRKCI and PRKCI mRNA in HPAEpiCs after treatment with ActD at the indicated time points. circ, circular RNA; PRKCI, protein kinase C iota; ALI, acute lung injury; qRT-PCR, quantitative real-time PCR; HPAEpiCs, human pulmonary alveolar epithelial cells; LPS, lipopolysaccharide; RNase R, Ribonuclease R; ActD, Actinomycin D. Data were presented as the mean ± SD, $n$ = 3. ns represent not significant, *$P < 0.05$. Power analysis: F1A: 0.94; F1B: 1; F1C-1: 0.38; F1C-2: 1; F1D:1.

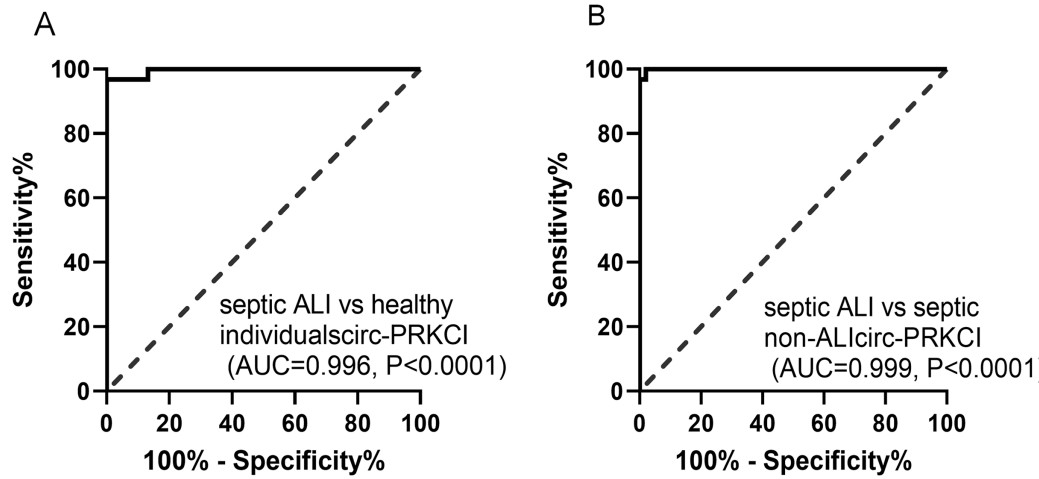

**Figure 2 The diagnostic utility of circ-PRKCI in sepsis-induced ALI.** (A) ROC curves analysis of circ-PRKCI for differentiating septic ALI patients from healthy individuals. (B) circ-PRKCI enabled high diagnostic capacity with an AUC value of 0.999 between septic ALI patients and septic non-ALI patients. ROC, receiver operating characteristic; AUC, area under the curve.

circ-PRKCI overexpression vector had no significant effects on PRKCI mRNA expression compared with the circ-NC group (Fig. 3A).

Firstly, CCK-8 assay showed that LPS treatment led to a significant decrease in the cell viability of HPAEpiCs, while overexpression of circ-PRKCI could reverse the LPS-induced the inhibition of cell viability (Fig. 3B). Then the levels of $Fe^{2+}$, GSH, MDA, and ROS were detected using corresponding commercial kits. Results revealed that LPS stimulation increased the levels of $Fe^{2+}$, ROS and MDA in HPAEpiCs, but introduction of circ-PRKCI significantly reduced $Fe^{2+}$, ROS and MDA production (Figs. 3C–3F). Meanwhile, LPS

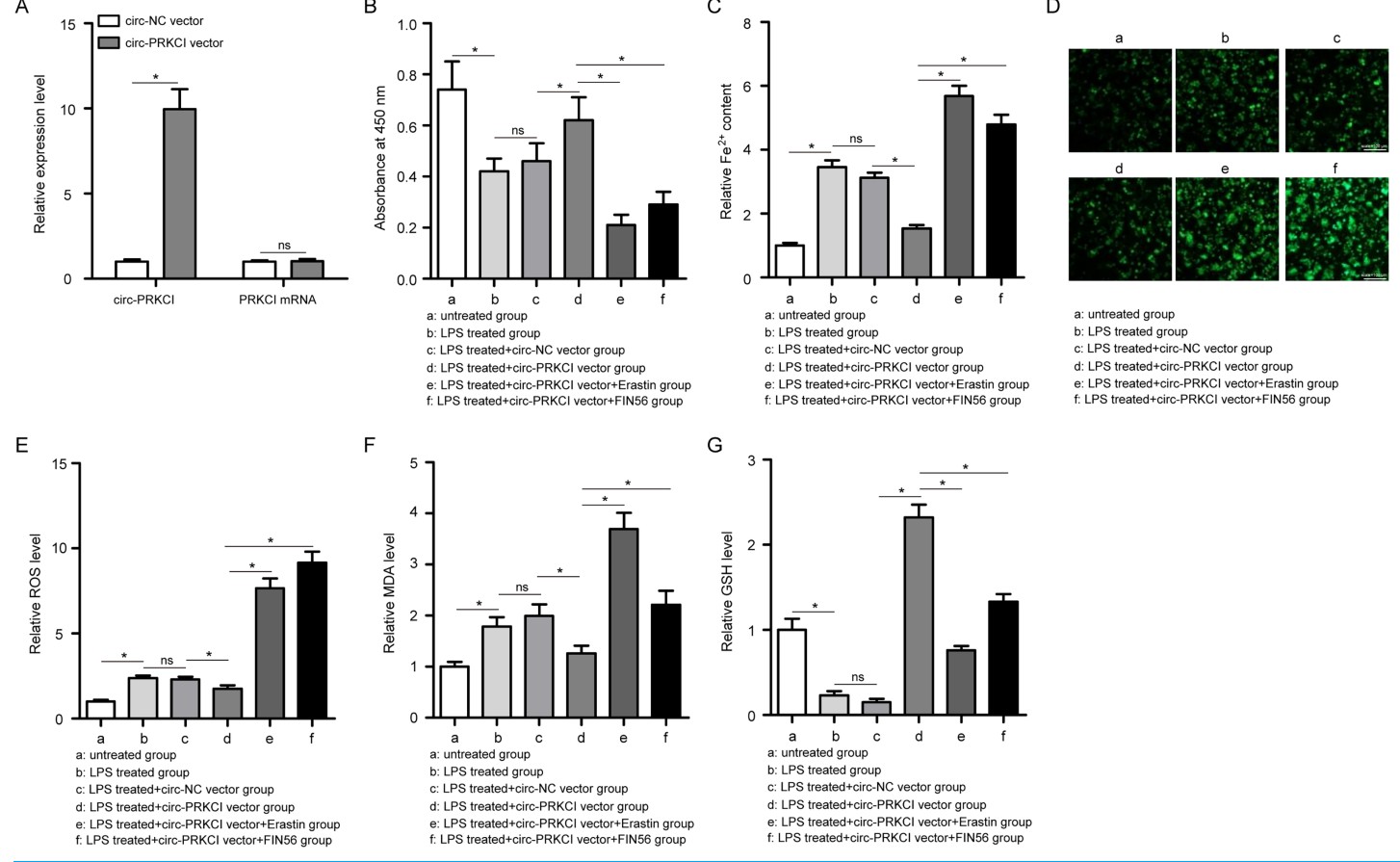

**Figure 3 The effect of circ-PRKCI on LPS-induced ferroptosis of HPAEpiCs.** (A) The measurement efficiency of circ-PRKCI overexpression in HPAEpiCs by qRT-PCR. (B) The analysis of cell viability in HPAEpiCs that underwent various treatments by CCK-8 assay. (C–F) The corresponding commercial kits analysis of $Fe^{2+}$, ROS and MDA levels in HPAEpiCs that underwent various treatments. (G) GSH assay was conducted to determine the GSH level in HPAEpiCs that underwent various treatments. CCK-8, Cell Counting Kit-8; $Fe^{2+}$, iron content; MDA, malondialdehyde; ROS, reactive oxygen species; GSH, glutathione. Data were presented as the mean ± SD, $n = 3$. ns represent not significant, *$P < 0.05$. Power analysis: F3A-1: 1; F3A-2: 0.11; F3B: 0.99; F3C: 0.99; F3E: 0.99; F3F: 0.99; F3G: 0.99.

exposure resulted in an inhibitory effect on GSH level, while the effect was reversed following the upregulation of circ-PRKCI (Fig. 3G). Moreover, the ELISA assay showed that LPS administration significantly increased the release of pro-inflammatory factors IL-6, IL-1β and TNF-α in HPAEpiCs, which was attenuated due to circ-PRKCI overexpression (Fig. 4). Additionally, the inhibitory effects of circ-PRKCI overexpression on ferroptosis and inflammatory response in LPS-induced HPAEpiCs were compromised after Erastin or FIN56 treatment. Therefore, we concluded that circ-PRKCI mitigates LPS-induced ferroptosis and inflammatory response of HPAEpiCs, suggesting that circ-PRKCI might play a protective role in sepsis-induced ALI.

## miR-382-5p is strikingly upregulated in sepsis-induced ALI and negative correlates with circ-PRKCI expression

To identify miRNAs regulated by circ-PRKCI, we performed bioinformatics analysis using CircInteractome and circBank1.0, and screened three candidate miRNAs including miR-

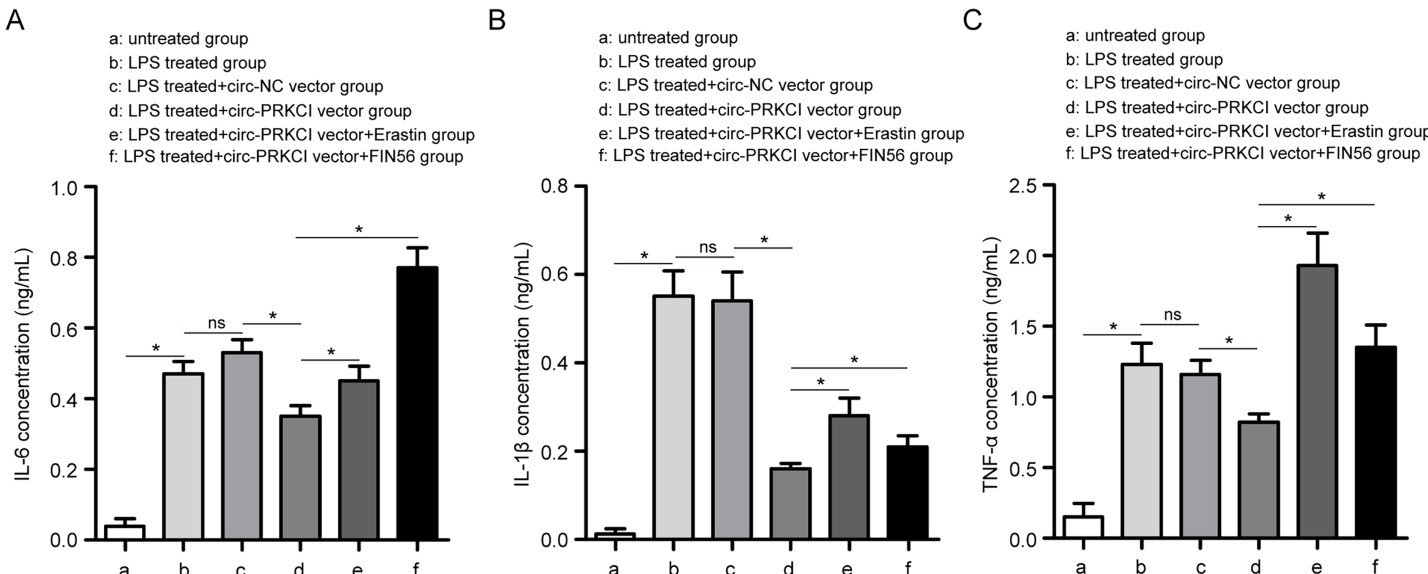

**Figure 4 The effect of circ-PRKCI on LPS-induced inflammatory response of HPAEpiCs.** ELISA assay was conducted to determine the release of pro-inflammatory factors IL-6 (A), IL-1β (B) and TNF-α (C) in HPAEpiCs that underwent various treatments. ELISA: enzyme-linked immuno-sorbent assay, IL-6, interleukin 6; IL-1β, interleukin 1 beta; TNF-α: tumor necrosis factor alpha. Data were presented as the mean ± SD, $n = 3$. ns represent not significant, *$P < 0.05$. Power analysis: F4A: 0.99; F4B: 0.99; F4C: 0.99.

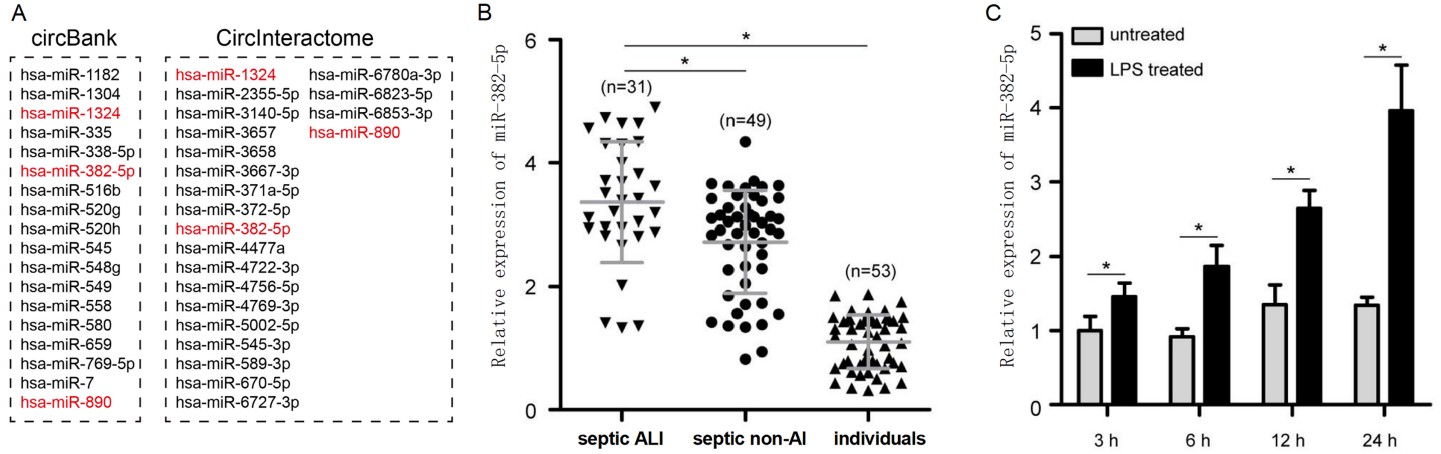

**Figure 5 The expression of miR-382-5p in sepsis-induced ALI and its correlation with circ-PRKCI.** (A) A schematic illustration demonstrating the putative miRNAs targeted by circ-PRKCI. (B) qRT-PCR detected miR-382-5p expression in 31 septic ALI patients, 49 septic non-ALI patients and 53 healthy individuals. (C) qRT-PCR analysis for the relative levels of miR-382-5p in HPAEpiCs treated with LPS for different times. miR, microRNA. Data were presented as the mean ± SD, $n = 3$. *$P < 0.05$. Power analysis: F5B: 1; F5C: 1.

1324, miR-382-5p, and miR-890 (Fig. 5A). Among these miRNAs, only miR-382-5p expression was significantly inhibited in HPAEpiCs by the overexpression of circ-PRKCI (Fig. S1). Moreover, qRT-PCR found that miR-382-5p levels were drastically upregulated in septic ALI patients, relative to healthy individuals and septic non-ALI patients (Fig. 5B). Similarly, miR-382-5p was highly expressed in LPS-induced HPAEpiCs (Fig. 5C). These results suggested a possible regulatory relationship between circ-PRKCI and miR-382-5p.

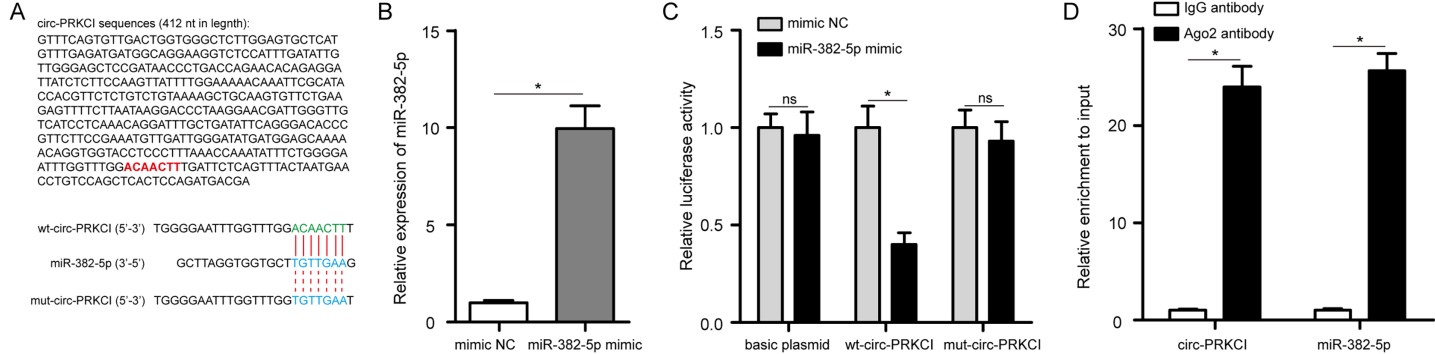

**Figure 6 circ-PRKCI functions as a sponge for miR-382-5p.** (A) The binding sites between circ-PRKCI and miR-382-5p were shown. (B) The measurement efficiency of miR-382-5p overexpression in HPAEpiCs after transfected with miR-382-5p mimic by qRT-PCR. (C) Measurement of relative luciferase activity in wt- circ-PRKCI plasmid or mut- circ-PRKCI plasmid or basic plasmid by luciferase reporter assays in HPAEpiCs. (D) The enrichment of circ-PRKCI and miR-382-5p in HPAEpiCs was evaluated by RIP assay using the Ago2 or IgG antibody. wt, wild-type; mut, mutant; RIP, RNA immunoprecipitation; Ago2, argonaute2; IgG, immunoglobulin G. Data were presented as the mean ± SD, $n$ = 3. ns represent not significant, *$P$ < 0.05. Power analysis: F6B: 1; F6C-1: 0.12; F6C-2: 1; F6C-3: 0.13; F6D-1: 1; F6D-2: 1.

## Circ-PRKCI serves as a ceRNA of miR-382-5p in HPAEpiCs

Circ-PRKCI sequences were noticed to contain the binding sites for miR-382-5p (Fig. 6A). Next, we conducted luciferase reporter assay with wt-circ-PRKCI plasmid or mut-circ-PRKCI plasmid and miR-382-5p mimic co-transfecting into HPAEpiCs. The efficiency of miR-382-5p mimic was validated in HPAEpiCs (Fig. 6B). As shown in Fig. 6C, miR-382-5p mimic significantly reduced the relative luciferase activity of wt-circ-PRKCI plasmid, but failed to reduce the relative luciferase activity of mut-circ-PRKCI plasmid, indicating that miR-382-5p could directly bind to circ-PRKCI. As expected, RIP analysis confirmed that circ-PRKCI and miR-382-5p were significantly enriched in the Ago2 antibody group, compared to the IgG antibody group (Fig. 6D). These results evidenced that circ-PRKCI functions as a sponge for miR-382-5p in HPAEpiCs.

## miR-382-5p abolishes the protective effect of circ-PRKCI on LPS-induced HPAEpiCs

Furthermore, to confirm whether the protective effect of circ-PRKCI on LPS-induced HPAEpiCs is targeted by miR-382-5p rescue experiments were performed with a circ-PRKCI overexpression vector together with a miR-382-5p mimic or mimic NC co-transfecting into HPAEpiCs and subsequently subjected to LPS treatment for 24 h. Interestingly, the CCK-8 assay exhibited that overexpression of circ-PRKCI and miR-382-5p partially attenuated cell viability compared with alone circ-PRKCI overexpression in HPAEpiCs (Fig. S2A). Moreover, miR-382-5p mimic could rescue circ-PRKCI overexpression-inhibited $Fe^{2+}$, ROS and MDA production in HPAEpiCs (Figs. S2B–S2E). In contrast, miR-382-5p mimic reversed circ-PRKCI overexpression-enhanced level of GSH in HPAEpiCs (Fig. S2F). In addition, ELISA data showed that the inhibition of IL-6, IL-1β and TNF-α by circ-PRKCI overexpression was partially reversed in HPAEpiCs after transfection of miR-382-5p mimic (Fig. S2G). Overall, our research demonstrated that

circ-PRKCI relieves ferroptosis and inflammatory response in sepsis-induced ALI of HPAEpiCs by sponging miR-382-5p.

## DISCUSSION

Sepsis is a lethal disease caused by imbalanced responses to infection, resulting in multiple organ damage or failure. Ferroptosis is a distinct form of programmed cell death with unique morphological and biochemical features, in which abnormal intracellular iron metabolism and lipid peroxidation are two indispensable hallmark processes. Growing studies have shown that ferroptosis plays a critical role in the development of sepsis-induced ALI (*Liu, Zhang & Xie, 2022*; *Wang, Zhao & Xiao, 2023*). It has been reported that Ferrostatin-1 alleviates LPS-induced ALI in human bronchial epithelial cell BEAS-2B *via* inhibiting ferroptosis (*Liu et al., 2020*). Panaxydol attenuates ferroptosis against LPS-induced ALI in mice *via* Keap1-Nrf2/HO-1 pathway (*Li et al., 2021*). STAT6 is a pivotal regulator of ferroptosis, which can inhibit ferroptosis and alleviate ALI *via* regulating P53/SLC7A11 pathway (*Yang et al., 2022*). Additionally, Itaconate inhibits ferroptosis of macrophage *via* Nrf2 pathways against sepsis-induced ALI (*He et al., 2022*). Nowadays, more and more circRNAs have been confirmed to be involved in ferroptosis of human diseases (*Yang et al., 2023*; *Zheng & Zhang, 2023*; *Zhou et al., 2023*), suggesting that ferroptosis-related circRNAs could serve as therapeutic targets. In the present study, we confirmed that circ-PRKCI is downregulated in sepsis-induced ALI. We firstly proved that circ-PRKCI could inhibit LPS-evoked ferroptosis and inflammatory response in HPAEpiCs, suggesting that circ-PRKCI may play a protective role in sepsis-induced ALI.

Several recent studies have investigated the roles of circ-PRKCI in various diseases, shedding light on its potential as a biomarker and a therapeutic target (*Qiu et al., 2024*). *Cheng et al. (2019)* demonstrated the involvement of circ-PRKCI in oxidative stress-injured neuronal cells, showing that overexpression of circPRKCI in SH-SY5Y cells significantly attenuated $H_2O_2$-induced cytotoxicity by regulating miR-545/589-E2F7 axis. *Zhou et al. (2018)* explored the role of circ-PRKCI in Hirschsprung disease, indicating that knockdown of circ-PRKCI suppressed cell proliferation and migration. *Chen et al. (2021)* found that circ-PRKCI targeted miR-1294 and miR-186-5p by downregulating FOXK1 expression to impede glycolysis in the progression of hepatocellular carcinoma. Although circ-PRKCI has been reported to relieve inflammatory injury in LPS-induced acute kidney injury by suppressing miR-545/ZEB2 (*Shi et al., 2020*), the functional role of circ-PRKCI in sepsis-induced ALI has not been previously reported. Herein, functional analysis suggested that circ-PRKCI protected HAPEpiC from the LPS-induced ferroptosis and inflammatory response, manifested as promoting cell viability, reducing $Fe^{2+}$, ROS and MDA production, increasing GSH level, and inhibiting the release of pro-inflammatory cytokines IL-6, IL-1β and TNF-α. These findings *in vitro* supported that circ-PRKCI impedes ferroptosis of sepsis-induced ALI. Similarly, a recent investigation revealed that knockdown of circ-EXOC5 reduces ferroptosis in sepsis-induced ALI (*Wang et al., 2022*).

Mounting evidence suggested that dysregulation of circRNAs is an early event in a wide range of diseases, including sepsis (*Wei et al., 2022*). Owing to their highly stable structural makeup, circRNAs are ideal diagnostic biomarkers for the blood testing-based evaluation

of patients in a minimally invasive manner for a range of diseases. *Peng et al. (2024)* found that circ_0072463 is a potential diagnostic marker for septic acute kidney injury. *Tian et al. (2021)* reported that circ_104484 and circ_104670 in serum have the potential to be used as diagnostic markers for sepsis. Our study revealed that the AUC values of the ROC curves for circ-PRKCI in differentiating septic ALI patients from healthy individuals and septic non-ALI patients were 0.996 and 0.999, suggesting circ-PRKCI has a high diagnostic potential as a biomarker for sepsis-induced ALI, which is in accordance with a previous study, that found that circ-PRKCI has high diagnostic capacity with an AUC value of 0.898 for sepsis (*Wei & Yu, 2020*).

Mounting evidence has suggested that circRNAs were involved in biological processes directly targeting miRNAs. In recent years, several miRNAs have also been identified as the downstream targets of circ-PRKCI, including miR-335 (*Liu et al., 2021*), miR-545-3p (*Qian et al., 2022*), and miR-3680-3p (*Shi et al., 2019*). According to the ceRNA hypothesis, our study confirmed that circ-PRKCI directly targeted miR-382-5p by luciferase reporter and RIP assays. miR-382-5p was identified to participate in various pathological processes across different diseases, especially in inflammation response. In breast cancer, *Ho et al. (2017)* demonstrated that miR-382-5p aggravates cancer progression by regulating the RERG/Ras/ERK signaling axis. *Yin et al. (2022)* reported that miR-382-5p inhibition hinders inflammatory response of IL-6 and TNF-α and neuronal apoptosis *via* targeting dual specificity phosphatase 1 (DUSP1). *Xiang et al. (2021)* elucidated the role of miR-382-5p overexpression in promoting LPS effect on the inflammatory factors of the microglia. Besides, in acute myocardial infarction, interference with miR-382-5p reduced apoptosis of myocardial cells (*Zhang et al., 2021b*). Previously, it has been reported that miR-382-5p promotes ferroptosis and represses progression of ovarian and breast cancer by targeting the SLC7A11 axis (*Sun, Li & Zhang, 2021*). Consistently, we observed that overexpression of miR-382-5p could reverse the protective effect of circ-PRKCI on LPS-induced HPAEpiCs. Based on these results, the present study demonstrated the innovative mechanism of circ-PRKCI in the regulation of ferroptosis and inflammatory response in sepsis-induced ALI by sponging miR-382-5p.

Several limitations of this study should be noted. First, as a retrospective study, bias in patients recruitment was inevitable, and the sample size of ALI patients in this study was relatively small, so further prospective studies are needed to collect more clinical samples to confirm the diagnostic value of circ-PRKCI in septic ALI. Second, the role of the circ-PRKCI/miR-382-5p axis in septic ALI has been demonstrated only at the *in vitro* cellular level, further work is needed to validate these finding *in-vivo*/animal models of septic ALI. In addition, the current study has not investigated the downstream mRNAs and related signaling pathways regulated by the circ-PRKCI/miR-382-5p axis, and we will investigate these mechanisms in future work.

## CONCLUSION

To sum up, our study demonstrated that circ-PRKCI is downregulated in sepsis-induced ALI and is a promising biomarker for septic ALI diagnosis. circ-PRKCI attenuates LPS-induced ALI by targeting miR-382-5p to inhibit ferroptosis and inflammatory

response. Our data unraveled that the circ-PRKCI/miR-382-5p axis may act as a potential therapeutic target for the prevention of sepsis-induced ALI.

### Funding
This work was supported by The Project of People's Hospital of Suzhou New District (SGY2022C01, SGY2023C02) and The Project of Suzhou (GSWS2023036). The funders had no role in study design, data collection and analysis, decision to publish, or preparation of the manuscript.

### Grant Disclosures
The following grant information was disclosed by the authors:
Project of People's Hospital of Suzhou New District: SGY2022C01, SGY2023C02.
Project of Suzhou: GSWS2023036.

### Competing Interests
The authors declare that they have no competing interests.

### Author Contributions
- Limei Yan conceived and designed the experiments, analyzed the data, authored or reviewed drafts of the article, and approved the final draft.
- Xiajun Lu performed the experiments, prepared figures and/or tables, and approved the final draft.
- Ning Wang performed the experiments, prepared figures and/or tables, and approved the final draft.
- Peng Jia conceived and designed the experiments, performed the experiments, analyzed the data, prepared figures and/or tables, authored or reviewed drafts of the article, and approved the final draft.

### Human Ethics
The following information was supplied relating to ethical approvals (*i.e.*, approving body and any reference numbers):
Medical Ethics Committee of People's Hospital of Suzhou New District (No. 2024-110).

### Data Availability
The raw data are available in the Supplemental File.

### Supplemental Information
Supplemental information for this article can be found online at http://dx.doi.org/10.7717/peerj.19404#supplemental-information.

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
