# Peer review of "Has_circRNA_0122683 (circ-PRKCI) relieves ferroptosis of HPAEpiCs in sepsis-induced acute lung injury by sponging miR-382-5p"

_PeerJ, doi:10.7717/peerj.19404_

## Round 0.1 · original submission · Minor Revisions

Please incorporate all reviewer comments into the revised manuscript and provide a detailed, point-by-point response in a separate document.

Reviewer 1 ·

Basic reporting

The research work titled “Has_circRNA_0122683 (circ-PRKCI) relieves ferroptosis of HPAEpiCs in sepsis-induced acute lung injury by sponging miR-382-5p” was performed and written very well. The authors have taken extensive efforts making the research happen. However, the following concerns can be addressed to make the manuscript appear better.
• The AUC values are promising. However, description related to ROC curves lacks clarity,
• Are there any bias identified during patient recruitment?
• Mentioning the study bias in the limitation section is better (related to recruitment, in-vivo/animal-based studies)
• The miRNA downstream process is not explained, these can be written in the future research recommendations
• Ethical section of the study is clear, However, adding more details on the informed consent procedure and sample storage conditions would strengthen the study ethically
• The study has stated the circ-PRKCI/miR-382-5p for their therapeutic implications. However, this was just a hypothesis. The study lacks in-vivo validation. This lacuna can be address in limitations of the study or can be mentioned in the future research prospects.
• The study is well justified as it addresses the role of circ-PRKCI in sepsis induced ALI, a condition with limited therapeutic targets. However, the novelty compared to previously published work on ferroptosis and circular RNA should be emphasized more clearly.
• Providing power analysis (statistically) would further strengthen the results.
• Relevant studies are cited. More recent publications would give better understanding various meta-analysis can be found online.
• The manuscript was identified to be having grammatical mistakes. Linguistic edition is recommended.

Experimental design

The research work titled “Has_circRNA_0122683 (circ-PRKCI) relieves ferroptosis of HPAEpiCs in sepsis-induced acute lung injury by sponging miR-382-5p” was performed and written very well. The authors have taken extensive efforts making the research happen. However, the following concerns can be addressed to make the manuscript appear better.
• The AUC values are promising. However, description related to ROC curves lacks clarity,
• Are there any bias identified during patient recruitment?
• Mentioning the study bias in the limitation section is better (related to recruitment, in-vivo/animal-based studies)
• The miRNA downstream process is not explained, these can be written in the future research recommendations
• Ethical section of the study is clear, However, adding more details on the informed consent procedure and sample storage conditions would strengthen the study ethically
• The study has stated the circ-PRKCI/miR-382-5p for their therapeutic implications. However, this was just a hypothesis. The study lacks in-vivo validation. This lacuna can be address in limitations of the study or can be mentioned in the future research prospects.
• The study is well justified as it addresses the role of circ-PRKCI in sepsis induced ALI, a condition with limited therapeutic targets. However, the novelty compared to previously published work on ferroptosis and circular RNA should be emphasized more clearly.
• Providing power analysis (statistically) would further strengthen the results.
• Relevant studies are cited. More recent publications would give better understanding various meta-analysis can be found online.
• The manuscript was identified to be having grammatical mistakes. Linguistic edition is recommended.

Validity of the findings

The research work titled “Has_circRNA_0122683 (circ-PRKCI) relieves ferroptosis of HPAEpiCs in sepsis-induced acute lung injury by sponging miR-382-5p” was performed and written very well. The authors have taken extensive efforts making the research happen. However, the following concerns can be addressed to make the manuscript appear better.
• The AUC values are promising. However, description related to ROC curves lacks clarity,
• Are there any bias identified during patient recruitment?
• Mentioning the study bias in the limitation section is better (related to recruitment, in-vivo/animal-based studies)
• The miRNA downstream process is not explained, these can be written in the future research recommendations
• Ethical section of the study is clear, However, adding more details on the informed consent procedure and sample storage conditions would strengthen the study ethically
• The study has stated the circ-PRKCI/miR-382-5p for their therapeutic implications. However, this was just a hypothesis. The study lacks in-vivo validation. This lacuna can be address in limitations of the study or can be mentioned in the future research prospects.
• The study is well justified as it addresses the role of circ-PRKCI in sepsis induced ALI, a condition with limited therapeutic targets. However, the novelty compared to previously published work on ferroptosis and circular RNA should be emphasized more clearly.
• Providing power analysis (statistically) would further strengthen the results.
• Relevant studies are cited. More recent publications would give better understanding various meta-analysis can be found online.
• The manuscript was identified to be having grammatical mistakes. Linguistic edition is recommended.

Additional comments

The research work titled “Has_circRNA_0122683 (circ-PRKCI) relieves ferroptosis of HPAEpiCs in sepsis-induced acute lung injury by sponging miR-382-5p” was performed and written very well. The authors have taken extensive efforts making the research happen. However, the following concerns can be addressed to make the manuscript appear better.
• The AUC values are promising. However, description related to ROC curves lacks clarity,
• Are there any bias identified during patient recruitment?
• Mentioning the study bias in the limitation section is better (related to recruitment, in-vivo/animal-based studies)
• The miRNA downstream process is not explained, these can be written in the future research recommendations
• Ethical section of the study is clear, However, adding more details on the informed consent procedure and sample storage conditions would strengthen the study ethically
• The study has stated the circ-PRKCI/miR-382-5p for their therapeutic implications. However, this was just a hypothesis. The study lacks in-vivo validation. This lacuna can be address in limitations of the study or can be mentioned in the future research prospects.
• The study is well justified as it addresses the role of circ-PRKCI in sepsis induced ALI, a condition with limited therapeutic targets. However, the novelty compared to previously published work on ferroptosis and circular RNA should be emphasized more clearly.
• Providing power analysis (statistically) would further strengthen the results.
• Relevant studies are cited. More recent publications would give better understanding various meta-analysis can be found online.
• The manuscript was identified to be having grammatical mistakes. Linguistic edition is recommended.

·

Basic reporting

Clear and unambiguous, professional English used throughout.
Literature references, sufficient field background/context provided.
Professional article structure, figures, tables. Raw data shared.

Experimental design

Research question well defined, relevant & meaningful. It is stated how research fills an identified knowledge gap.
Rigorous investigation performed to a high technical & ethical standard.
Methods described with sufficient detail & information to replicate.

Validity of the findings

Impact and novelty not assessed. Meaningful replication encouraged where rationale & benefit to literature is clearly stated.
All underlying data have been provided; they are robust, statistically sound, & controlled.
Conclusions are well stated, linked to original research question & limited to supporting results.

Additional comments

the manuscript is clearly written in professional, unambiguous language. If there is an error, it is:
1- In the abstract you write
The area under the curve values of ROC curves for circ-PRKCI in differentiating septic ALI patients from healthy individuals and septic non-ALI patients were 0.948 and 0.814. While in Figure 2 The diagnostic utility of circ-PRKCI in sepsis-induced ALI. AUC value was 0.996 and 0.999
2- In the supplementary figures
In the main manuscript you write Supplementary Figure 1. Effect of circ-PRKCI on the expression of miR-382-5p. qRT-PCR was performed for miR-382-5p expression detection after circ-PRKCI overexpression in HPAEpiC. Data were presented as the mean ± SD, n = 3. ns represent not significant, *P<0.05.
While in raw data and figure 1 in the supplementary data show the expression of miR-890, miR-1324 and miR-382-5p in circ-NC vector (A group) and circ-PRKCI vector (B group)

---

## Round 0.2 · accepted · Accept

Authors have addressed all of the reviewers' comments and the manuscript is ready for publication.

·

Basic reporting

Clear and unambiguous, professional English used throughout.

Literature references, sufficient field background/context provided.

Professional article structure, figures, tables. Raw data shared.

Experimental design

Original primary research within Aims and Scope of the journal.
Research question well defined, relevant & meaningful. It is stated how research fills an identified knowledge gap.

Validity of the findings

Meaningful replication encouraged where rationale & benefit to literature is clearly stated.
Conclusions are well stated, linked to original research question & limited to supporting results.